# An Update of Moisture Barrier Coating for Drug Delivery

**DOI:** 10.3390/pharmaceutics11090436

**Published:** 2019-09-01

**Authors:** Qingliang Yang, Feng Yuan, Lei Xu, Qinying Yan, Yan Yang, Danjun Wu, Fangyuan Guo, Gensheng Yang

**Affiliations:** 1College of Pharmaceutical Science, Zhejiang University of Technology, Hangzhou 310014, China; 2Research Institute of Pharmaceutical Particle Technology, Zhejiang University of Technology, Hangzhou 310014, China

**Keywords:** moisture barrier coating, hydrolytic degradation, coating formulation, coating method, dry coating

## Abstract

Drug hydrolytic degradation, caused by atmospheric and inherent humidity, significantly reduces the therapeutic effect of pharmaceutical solid dosages. Moisture barrier film coating is one of the most appropriate and effective approaches to protect the active pharmaceutical ingredients (API) from hydrolytic degradation during the manufacturing process and storage. Coating formulation design and process control are the two most commonly used strategies to reduce water vapor permeability to achieve the moisture barrier function. The principles of formulation development include designing a coating formulation with non-hygroscopic/low water activity excipients, and formulating the film-forming polymers with the least amount of inherent moisture. The coating process involves spraying organic or aqueous coating solutions made of natural or synthetic polymers onto the surface of the dosage cores in a drum or a fluid bed coater. However, the aqueous coating process needs to be carefully controlled to prevent hydrolytic degradation of the drug due to the presence of water during the coating process. Recently, different strategies have been designed and developed to effectively decrease water vapor permeability and improve the moisture barrier function of the film. Those strategies include newly designed coating formulations containing polymers with optimized functionality of moisture barrier, and newly developed dry coating processes that eliminate the usage of organic solvent and water, and could potentially replace the current solvent and aqueous coatings. This review aims to summarize the recent advances and updates in moisture barrier coatings.

## 1. Introduction

The stability of an active pharmaceutical ingredient (API) in dosage form during its shelf life is essential to guaranteeing its efficacy and safety. Degradation of the API occurs through chemical reactions, including oxidation, photo and thermal degradation, and hydrolysis, which makes the API unstable, hence significantly hindering its therapeutic effect [1]. Among these various processes, hydrolytic degradation is most commonly found to reduce the stability and bioavailability of the APIs specifically for moisture-sensitive drugs, including many herbal extracts such as traditional Chinese medicines. Although the inherent humidity of the core excipients needs to be carefully considered and controlled in the formulations [2], atmospheric moisture remains the predominant source of water that chemically or physically impact the active ingredients.

Several approaches have been developed to minimize water uptake into dosage cores, thus preventing the hydrolytic degradation of the active ingredients. These approaches include formulation designs with appropriate excipients [3] that repel water, avoiding ingredients with high inherent moisture, packaging the dosages with suitable materials and adding desiccants [4,5,6], and coating the dosage cores with polymers to achieve a moisture barrier film [7]. Choosing appropriate excipients and packaging materials requires little production time compared to film coating, however, they offer limited alternatives, particularly for drug delivery. Therefore, the most appropriate and widely used approach is to protect the cores with a moisture barrier film that could effectively prevent water vapor from reaching the cores, thus eliminating the interaction between moisture and the active drug substances that are susceptible to hydrolysis. Film coating also protects the cores from the external environment blocking light and hence preventing oxidation as well as modifying the drug release characteristics [8].

An ideal film coating should exhibit several qualities to achieve the function of a moisture barrier. First, the coating film should have an adequate thickness and density, with a low permeability to water vapor. Such a coating should also possess high uniformity both in thickness and density, with as few defects as possible, reducing the possibilities for water vapor transmission. It is also important for the coating film to be stable during the dosage shelf life without any time-dependency or moisture stress-induced relaxation and ageing [9], which could possibly promote water vapor transmission through the film.

In the past several decades, polymeric film coating has transited from organic solvent coating to aqueous coating, driven by the increasing strictness in safety and environmental requirements. For organic solvent coating, the coating polymer is firstly dissolved in an appropriate solvent to form a coating solution, followed by a spraying process to obtain a coating film. The process for aqueous coating is more complicated, due to the differences in water-solubility of the coating polymers. For water-soluble polymers, the process is similar to organic solvent coating. However, for water-insoluble coating polymers, instead of a coating solution, it is rather a coating suspension formed by dispersing of the micronized particles of coating polymers into water. Compared to aqueous coating, organic solvent coating is advantageous in producing a faster and more uniform coating film owing to the dissolved nature of the coating polymers. However, it has been phased out by aqueous coating in the 1990s because of toxicity and environmental issues. Although aqueous coating started to dominate and remains the preferred coating method in the pharmaceutical industry, it still has many limitations [10], such as a longer processing time and a higher energy consumption. Most importantly, aqueous coating is not appropriate for those moisture-sensitive APIs.

To overcome these limitations, many alternative coating techniques [11,12,13] have been designed and developed in recent years, in which the use of organic solvent and water is reduced or avoided. In addition, several coating formulations have also been developed to achieve a better moisture barrier function.

This review details the mechanism of moisture uptake through the coating film, summarizes newly developed coating methods and formulations with their respective advantages and limitations, and provides a brief future perspective for applying those new technologies and formulations in the pharmaceutical industry.

## 2. Principles of Moisture Uptake

### 2.1. Mechanism of Moisture Uptake

Conceptually, the mechanism of moisture passage through a film to the surface of the drug substrate [14] involves three steps, as shown in Figure 1. First, water molecules migrate to, and are adsorbed by, the film’s surface. The molecules then diffuse through the insulating film, followed by the desorption of the water molecules by the film–substrate interface. The cycle continues until an equilibrium is achieved.

The relationship between water vapor diffusion and the film can be described by Crank’s equation [15]:
*P* = *D* × *S*(1)
where *P* is the coefficient of permeability, D is the coefficient of diffusion, and S is the solubility. As can be seen, the diffusion of water through the film is dependent on the water-solubility of the film and permeability of the film. Essentially, water vapor permeability is associated with the relative polarity of the polymer, where the coefficient of permeability of the film increases as the structural similarities between the polymer and the diffusing molecule increase. Furthermore, the speed and intensity of the diffusion is determined by the permeability of the film and, thus, the rate of diffusion is governed by the coefficient of diffusion of water in that polymer.

In addition, water sorption frequently exhibits hysteresis, which generally occurs when adsorption happens by merging clumps of molecules on separate sites. Desorption hysteresis is caused by a low vapor pressure led by all the molecules that are held by the forces from all active sites touched by the merged clumps, and the force is stronger than when the clumps are separated.

### 2.2. Testing of Moisture Uptake

A number of methods [16] have been developed to test the moisture uptake of pharmaceutical dosages. The most commonly used approach is to measure the weight increase of the dosages at various constant temperature and humidity conditions. Typically, both coated and uncoated dosages are evaluated for their moisture uptake at these different humidity conditions created by a saturated salt solution, such as potassium chloride or sodium chloride. According to the United States Pharmacopeia (USP), for equilibrium moisture determinations, weighing should be carried out every hour until achievement of consecutive readings corresponding to a recorded mass change of less than 0.25%.

Another widely used testing method is the water vapor permeability test (WVP) [17], where the amount of water that permeates through the film per unit area is evaluated over time (kg/m s Pa). As an indicator of the moisture-proofing ability of the coating film, most of the WVP experimental data is obtained by following the American Society for Testing and Materials (ASTM) Standard Test Method E96/E96M. According to this standard, the coating film is prepared with certain polymers, and the thickness of the prepared film is measured with a micrometer. Then, the prepared film is sealed in a cup and placed under a controlled temperature and humidity. Finally, WVP is achieved by measuring the amount of water that penetrated through the film.

Thermal analysis techniques, including differential scanning calorimetry (DSC), and thermogravimetric analysis (TGA) [18], could also be utilized to estimate the moisture resistance capacity of the coating film after being stored at elevated temperatures and humidity conditions by evaluating the reduction in the glass transition temperature (*T*_g_) of the polymer film due to the plasticizing function of water in the polymeric film [19]. The *T*_g_ is defined as the temperature at which polymeric materials in the glassy state transform into a supercooled liquid state upon heating, resulting in the rearrangement of the molecules into a new, higher energetic structure. The *T*_g_ of the pharmaceutical polymers can be adjusted by the plasticizing effect with specific plasticizers, which dramatically affects the moisture protection ability and the stability of the formulation. Water is a good plasticizer for most of the coating polymers, indicating that the more moisture permeated through the film, the lower of the *T*_g_ will be due to the plasticizing effect. For the TGA method, the higher weight loss at the temperature of 100 °C demonstrates that more moisture permeated through the coating film, indicating weaker moisture resistance.

In addition, moisture uptake could also be tested by precisely measuring the mass change of the samples. One of the most widely used techniques is the dynamic vapor sorption technique (DVST) [20]. In this system, the material sample is suspended using an ultrasensitive microbalance that is capable of measuring the very small mass changes of both the uptake and loss of moisture caused by the flow of a carrier gas at a specific relative humidity (RH). Compared to the traditional approaches using salt solutions, DVST is able to provide better measurements with high precision/sensitivity in mass determination.

## 3. Formulations of Moisture Barrier Coating

A dosage’s coating film plays a crucial role in determining its moisture barrier property. Consequently, the most important factor that influences moisture uptake is the water vapor permeability of the coating film. It is related to the film’s thickness and mechanical properties, as well as its physical and chemical stabilities, all of which are depended on the coating film formulation, composed of film-forming polymers [21] and other necessary ingredients [22] such as plasticizers and pigments.

The thickness of the coating film remains the key factor when it comes to the film’s resistance to moisture. Increasing the thickness [23,24] prolongs the disintegration time and improves the tensile strength of the core. In a typical liquid coating process, a larger amount of moisture could be retained in the film by increasing its thickness. In addition to the thickness, a film’s uniformity also influences its moisture absorption since varying thickness levels will retain different amounts of moisture. Moreover, thinner films could result in a faster drug release and instant therapeutic effects compared to thicker films. Appropriate coating levels must thus be developed to provide efficient and reliable products.

### 3.1. Film-Forming Polymers

Most of the film coating materials used for moisture protection are synthetic polymers, which are categorized as water-soluble polymers, water-insoluble polymers, and entero-soluble polymers.

Water-soluble polymers are widely employed in moisture barrier coating (Table 1), including polyvinyl alcohol (PVA) [25], hydroxypropyl methyl cellulose (HPMC), hydroxyethyl cellulose (HEC), and polyvinyl alcohol-polyethylene glycol (PVA–PEG copolymer). The water-solubility of these polymers makes them the preferred materials for moisture-protective coating as they do not influence drug release or the therapeutic effect. They can also easily be used in the aqueous coating process, where the coating polymers are dissolved in water to form a coating solution, eliminating the issues related to the use of organic solvent coating. Besides moisture protection, some of the water-soluble polymers could also be used to achieve a taste-masking coating. However, the coating film formed by the water-soluble polymers has a relatively shorter lifetime compared to the ones formed by the water-insoluble polymers owing to the degradation of the coatings caused by the ambient humidity during storage.

Water-insoluble polymers are mainly used as coating materials to modify and extend drug release to accomplish sustained or controlled release. Some of them can form a coating film with a low permeability and, thus, could also be used as moisture protection coating materials. Those polymers (Table 1) include cellulose esters, such as ethyl cellulose (EC) [26,27] and cellulose acetate (CA), and acrylic esters, such as ethyl acrylate–methyl methacrylate copolymers. The coating process consists of first spraying an aqueous coating suspension composed of fine particles of film-forming polymers as well as other excipients, such as plasticizers and pigments, onto the surface of the dosage cores, followed by a curing step to allow the deposited particles to coalesce and form the coating film.

A coating film formed with entero-soluble polymers (Table 1), such as shellac [28], and anionic copolymers based on methacrylic acid and methyl methacrylate (Eudragit^®^ L) [29] could efficiently provide both moisture protection and enteric functionalities [30] due to their water insolubility at neutral and acidic pH. The physicochemical properties of the coating film, including water vapor permeability and stability, are extensively attributed to the aggregated structure of the molecular chains of these polymers. However, both types of water-insoluble polymers and entero-soluble polymers need to be optimized by adding extra excipients or by controlling the thickness of the film to modify its solubility [31].

### 3.2. Plasticizers

Plasticizers [32] can change the physical properties of polymers by reducing the *T*_g_ of the film. The mechanism of action of plasticizers is generally based on the molecular interactions of polymer chains where plasticizers help polymer chains move and interact with each other to develop firmer bonds, thus allowing the molecules to coalesce and the film to form. Additionally, plasticizers help enhance the film’s adhesion to the tablet’s surface. However, water uptake can be thought of as having “dissolved” into the amorphous structure and acts as a plasticizer by promoting the glass-to-rubber state transition of the polymeric system [33,34]. In the presence of moisture, unplasticized films can also cause significant changes in the mechanical properties of the film. Therefore, by increasing the water content, the *T*_g_ of the polymeric system will decrease, which affects its moisture protection ability.

Nevertheless, appropriate concentrations and types of plasticizer play an important role in moisture-resistant film coatings. Above a certain concentration, molecular scale holes are generated, thereby increasing the speed and intensity of the diffusion of water molecules, resulting in a film with poor moisture protection. Moreover, the nature of the plasticizer drastically influences the moisture protection properties of the final film—hydrophilic plasticizers will modify the barrier properties and increase the water vapor permeability of the film, whereas hydrophobic plasticizers will decrease the film’s water uptake [35].

### 3.3. Pigments

Pigments [36,37] also play an important role in moisture-protective polymer coatings. There is a risk of intermolecular spaces in films as the formation of a complete barrier against moisture is impossible. Therefore, pigments, acting as insoluble additives, are used on the coatings to block these intermolecular spaces. At low critical pigment concentrations [38], water vapor permeability decreases as the pigment’s concentration increases. This is because the pigments are discrete particles that serve as barriers to the diffusion of moisture through the film. As the pigment concentration increases, its barrier efficacy will also increase until there are no polymers left to bind with. However, a high concentration of pigments may result in pores in the coating film, leading to an increased water vapor permeability. Hsu et al. [39] reported that low levels (10% (*w*/*w*)) of the whitener titanium dioxide (TiO_2_) slightly increased the water vapor permeability of polyvinyl alcohol film, but was followed by a sharp increase of permeability at high levels (20% (*w*/*w*)). Thus, the pigment concentration needs to be carefully optimized to ensure the moisture protection property of the final film. Moreover, the pigments need to be distributed uniformly in the coating formulation to prevent agglomeration, which could possibly result in a weaker film and an inelegant appearance. Traditionally, high-shear mixing equipment is utilized to achieve an adequate dispersion of pigments in the coating solution or coating suspension before the coating process. During coating, it is also necessary to ensure a uniform distribution by continuous low-shear stirring of the coating solution or dispersion.

A further coating formulation design alternative is to combine different polymers in the coating, preferably a combination of water-soluble and insoluble polymers at various ratios. As a result, the positive effects of both approaches will be combined, e.g., blocking the bitter-tasting molecules from interacting with the receptors on the tongue and instant drug release in the stomach.

## 4. Coating Process

Film coating in the pharmaceutical industry dates to the mid-1900s, when liquid-coated tablets were first introduced. The coating process [40,41] consists of preparing a coating solution or suspension by mixing a coating polymer with an appropriate liquid (either organic solvent or water) as well as some additives (such as plasticizers), and applying the coating solution or suspension to a solid dosage to control the drug release and protect the API against external elements. Stronger bonding between the polymer and the solvent will produce a more uniform coating. There are two types of liquid coating technologies: aqueous coating and organic solvent coating. Their use depends on several factors, including the polymer’s characteristics, product applications, environmental considerations, and productivity requirements, although aqueous coating is more widely used as there are some safety issues related to organic solvent coating.

### 4.1. Organic Solvent Coating

Although organic solvent coating is not the most widely used coating technology since the 1990s, due to toxicity and environment-related issues, it still remains the preferred coating method for special dosages, such as osmotic drug delivery systems [42], and for some specific drugs, especially moisture-sensitive drugs. For those special dosages, including osmotic drug delivery systems, the coating polymers generally have a very high *T*_g_ (cellulose acetate [43], *T*_g_ ≈ 185 °C), which is unacceptable for the aqueous coating to form a uniform and continuous coating film. Additionally, for those moisture-sensitive drugs, the humidity during the aqueous coating process will cause serious drug degradation.

Commonly used solvent materials include ethanol and isopropanol, which can also be mixed with acetone as they evaporate quickly and efficiently. Figure 2 shows a schematic illustration of the film’s formation process in an organic solvent coating process. Firstly, polymers, pigments, and excipients are mixed together and dissolved into an appropriate organic solvent to form the coating solution. Then, the polymeric system (coating solution) is sprayed onto the surface of the dosage core via an atomizing nozzle, following by a heating process to allow the organic compound to evaporate and the polymer particles to coalesce into a continuous film. Basically, an additional post-coating heating treatment is applied to ensure the removal of residual solvent and prolong the curing of the film. This results in a uniform, smooth, and lustrous coating surface. The organic solvent coating process is carried out either in a perforated pan coater for larger solid dosages, such as tablets and capsules, or in a fluidized bed coater for smaller dosages, including pellets, pills, and particles.

The rate of evaporation of the solvent is crucial in the film formation process. Below the optimum rate, the drug core will become sodden, especially for moisture-sensitive drugs, and in extreme cases, will start to dissolve. Thus, when the solvent evaporates too slowly, it will weaken the moisture-protective ability of the film. However, if the evaporation was too quick, film formation might have happened before polymer-containing droplets either adhere to the substrate’s surface or spread onto the surface due to an orange peel effect, leading to interrupted film formation and resulting in a weaker film. Weaker films have poor moisture-protective abilities as they allow moisture to move with more ease. Many variables affect the rate of evaporation of solvents, including temperature, atmospheric pressure, and air movement. Balancing these variables to produce a flexible, dry coating of adequate thickness is difficult, but can be achieved through careful manipulations of the coating process, which causes additional costs.

Generally, organic solvents are necessary to dissolve most sustained-release [39,40] and moisture-protective coating polymers due to their nature. These polymers contain either a hydrophobic or a lipophilic substance, i.e., shellac, zein, EC, an enteric polymer such as cellulose acetate phthalate (CAP) and similar polymers, or a fatty acid. A highly hydrophobic polymer reduces the water vapor permeability of the final film by preventing the movement of water molecules [44], thus providing moisture protection. EC is insoluble in water and has a low permeability and is thus used as a controlled release coating material in pharmaceutical applications. It also has potential as a moisture-protective coating and can extend the drug release through modification of the film’s thickness.

Some entero-soluble polymers, such as shellac, are also moisture-protective due to their low water permeability. Pearnchob [45] used shellac to coat acetylsalicylic acid tablets, and compared this with tablets coated with an aqueous solution of HPMC, the most frequently used coating formulation for moisture protection. The coating formulation composed of an ethanolic solution of shellac (10% (*w*/*v*), based on total solution) with triethyl citrate as plasticizer. Their results indicated that the shellac-coated tablets had a lower water uptake rate and a higher stability compared to the HPMC-coated systems with the same coating level. In addition, the inherent disadvantage of using an HPMC solution is its high energy consumption, which requires elevated temperatures during the coating process.

There are also some reported studies that tried to use plant-based materials to form the coating formulation. Hydrogenated rosin (HR) [46] is poorly soluble in water but soluble in organic solvents such as dichloromethane and chloroform, which makes it an efficient barrier against moisture migration when combined with a hydrophobic plasticizer such as dibutyl sebacate (DBS). Similarly, zein [47,48], one of the most commonly used biomaterials for coating water-soluble drugs such as metoprolol tartrate, can be used either with an aqueous dispersion or an organic solvent system (ethanol) to form biodegradable films that are very efficient moisture barriers. Studies on the water vapor barrier properties of coating systems reported no significant difference between aqueous films and organic solvent-based films.

### 4.2. Aqueous Polymeric Film Coatings

Although still in use today, organic solvent coating has, for the most part, been replaced by aqueous polymeric film coating in the pharmaceutical industry. Aqueous polymeric film coating is an alternative coating approach that avoids various disadvantages, such as toxicity, associated with residual organic solvents in the film and environmental issues such as producing large amounts of volatile organic compounds (VOCs).

Aqueous coating can be divided into two different situations according to the water-solubility of the coating polymers. The whole process is similar to organic solvent coating for water-soluble polymers, while for water-insoluble polymers, the coating process and film formation are different [49]. Water droplets are first created at the spraying nozzle, then atomized and accelerated toward the surface of the substrate while polymer spheres in the nanometer size range are dispersed and deposited over the same surface. Then, the substrate is heated to remove water and allow the polymer spheres to amalgamate. Finally, with additional heat, the polymer chains deform, interpenetrate, and coalescence to form the coating film.

The dry rate plays a critical role in achieving an acceptable coating film [50]. A rapid loss of water may not allow for the development of the necessary capillary force between the deposited coating particles and the surface of the cores, thus inhibiting the deformation and coalescence of the polymer particles, thereby resulting in a weaker moisture-proof film. By contrast, excess drying can prevent the aqueous coating droplets from spreading over the surface of the cores during the coating process, leading to a non-uniform distribution of coating polymers. Using aqueous polymeric systems [51] with a higher solid content (above 12% (*w*/*w*)) has also been suggested to ensure a lower humidity level during coating. However, a thick coating dispersion may easily block the spray nozzle.

Coating formulation design is based on various polymer types. Bley [52] evaluated the protective ability of different coating polymers with respect to coating and curing conditions. Tablets containing freeze-dried garlic powder, a moisture-sensitive herbal material, were coated and the water uptake rates were determined. The results show that tablets coated with polyvinyl alcohol and poly(methacrylatemethylmethacrylates) had the lowest moisture uptake rates with a significantly improved drug stability. Interestinglyonce the coating was obtained, curing at elevated temperature did not improve the moisture-protective ability of the polymeric film.

In addition, Bley [53] reported which type of polymers strongly affected water transport in the films when stored at room temperature and elevated relative humidity. This was analyzed using differential scanning calorimetry (DSC) and dynamic vapor sorption (DVS). The study shows that the glassy-to-rubbery state transition occurred for the polymeric system of Opadry^®^ AMB, while Eudragit^®^ E PO remained unchanged physically. Due to the higher mobility of its macromolecules, Opadry^®^ AMB-based films had a higher water uptake rate.

The efficacy of the moisture barrier for non-hygroscopic tablet cores using aspirin as the moisture-sensitive active ingredient and coated with aqueous dispersions of Eudragit^®^ L30 D-55, Eudragit^®^ E PO, Opadry^®^ AMB, and Sepifilm^TM^ LP, at the vendors’ recommended weight gains, were studied. However, the results show that the mean water uptake was higher for coated cores compared to uncoated ones [54]. Therefore, coated tablets had high water uptake rates, which accelerate drug degradation and, thus, the efficacy of polymer coating on low hygroscopic cores was limited.

Films made from highly hydrophilic polymers, which can be combined with a lipophilic substance, have potential moisture-protective properties. They prevent the penetration of moisture into the core by forming hydrogen bonds with water molecules. Penhasi [55] designed and developed a hybrid coating formulation for moisture barrier based on a solid dispersion of stearic acid (SA) in HPC using a polymeric surface-active agent (PSAA). They found that the formulation ratio amongst the different compositions could be optimized to considerably increase the barrier capability of the film against moisture, which is much better than the films formed by either pure HPC or the combination of HPC and SA. Additionally, these systems generally retain the basic properties of HPC.

In the pharmaceutical industry, there is increasing interest in the development of biopolymers as functional coatings. Abietic acid [56], an extracted product from rosin, was used to impart hydrophobicity and acid-resistant properties to provide the function of moisture protection. Similarly, zein, a cheap and readily available biomaterial, also has hydrophobic properties. Li [57] used a polyethylene glycol (PEG 400) plasticized pseudolatex film made of zein to coat metoprolol tartrate. The results showed that the moisture-resistant film had a low water absorption rate at high humidity conditions for very water-soluble drugs.

A combination of different polymers with various coating ratios has been shown to be a promising method to provide moisture protection, as some studies have established that moisture-protective properties can be modified by combining insoluble type polymers. Heinämäki [58] assessed the addition of suberin fatty acids (SFAs) into HPMC aqueous solution plasticized with PEG 400. They found that the water vapor barrier property was significantly increased with an increase of SFA concentration up to 15%. Lopez [59] used EC and pectin as film coating materials for pellets, with DBS as the plasticizer. The results showed that WVP values of the film were significantly affected by EC and DBS as they increased the water vapor permeability of the membrane. Notably, adding plasticizers into composite films can enhance its mechanical characteristics, whereas moisture protection is mainly provided by the polymers pectin and shellac [60]. The results indicated that only the addition of proper plasticizers can improve a film’s moisture-protective properties.

### 4.3. Dry Coatings

Over the past several decades, cost and environmental concerns associated with the use of solvents, such as ethanol, were minimized by using dry coatings, including compression coating, hot-melt coating, heat dry coating, electrostatic dry powder coating, and vapor phase deposition methods. These approaches are highly valued as they are safe, environmentally friendly, and have a lower overall cost due to the absence of organic solvents and water.

Compression coating [61,62] is a coating technique well suited for moisture- and heat-sensitive drugs. It is the simplest way to coat tablets as the coating materials are directly applied onto the cores through compression. However, this method creates non-uniform coating films due to the off-center positioning of the cores. To overcome this, Ozeki [63] designed a one-step dry-coated (OSDRC) tablet manufacturing method, which consists of an upper-center punch, a lower-center punch, an upper-outer punch, and a lower-outer punch that makes dry-coated tablets in a single run; thus, prefabrication of the tablet core is no longer necessary. Results have shown that crystallized amorphous sucrose, after being compressed at 200 MPa, becomes moisture-protective. Additionally, it was found that amorphous sucrose has better moisture resistance compared to HPMC when stored at 25 °C and 75% RH. Therefore, compression coating produces moisture-proof coated tablets.

Hot-melt coating requires a number of different lipid excipients as coating materials such as lipidic/waxy excipients with low melting points [64]. In general, a suitable material for hot-melt coating is one having a melting point within the range of 50–100 °C. These materials are heated to a molten state to produce a solution which is air-atomized and sprayed onto the surface of the solid dosage forms. Then, the dosages are cooled to form a continuous film that acts as a rate-controlling membrane for both drug release and moisture protection. A perforated pan coater and a fluid or spouted bed are needed for this process. This technology has been applied to Guizhi Fuling (GF) [65], an herbal compound, since it is highly hygroscopic. The results show that hot-melt coating forms better moisture-proofing films compared to aqueous coating, due to the methods’ varying moisture sorption mechanism. For hot-melt coating, the moisture sorption behavior is attributed to the water vapor diffusion via a porous matrix system which can be fitted to the Higuchi model. However, this coating method is only suitable for drugs with thermally stable properties, otherwise the drug would degrade during the coating process.

Cerea et al. [66] developed a heat dry coating technique, which applied a laboratory scale spheronizer to produce dry powder-coated theophylline tablets. The dry coating process involved a feed of solid coating materials into the spheronizer with a smooth stainless-steel disk. The edges of the disk are tilted at a 45° angle to facilitate the tumbling movement of the tablets and to prevent the loss of coating powder. Unlike other dry coating methods, this one requires the use of an infrared lamp with a variable transformer as the heating source, maintained at temperatures above the melting point of the polymeric system. Additionally, pipeline insulation was utilized to prevent the solidification of the molten materials before the final cooling takes place on the surface of the coating cores. With the micronized acrylic polymer Eudragit^®^ E PO as the coating material, this dry coating process was capable of being an alternative to solvent or aqueous film coating technologies for applying moisture-protective film coats onto compressed tablets.

Despite the advantages of the dry coating technologies as mentioned above, the deficient uniformity of the coating film is still the biggest challenge hindering their application and commercialization. Zhu et al. [67,68,69,70] developed an electrostatic dry coating method utilizing electrostatic deposition of the charged coating particles onto the surface of the dosages, which dramatically enhanced the uniformity of the coating film. Basically, the electrostatic coating process consists of two steps: coating particle deposition and film formation. The coating particles are first sprayed onto the surface of the dosages using an electrostatic spray gun to achieve adequate coating powder deposition. This process can be promoted by spraying a suitable amount of liquid plasticizer, which is capable of both increasing the electrical conductivity of the dosages and reducing the *T*_g_ of the coating polymer [71]. After the coating particle deposition, there is a curing step to allow those deposited coating particles to coalesce together to form a continuous coating film. As shown in Figure 3, the coating system [72,73,74] requires an electrically grounded coating pan, a charging gun, and a heating source. Normally, a liquid plasticizer spray gun is also needed to apply plasticizers to promote both the coating powder adhesion and film formation under a lower curing temperature with a shorter processing time [75,76]. Qiao [77] applied this electrostatic dry coating process to coat ibuprofen tablets with ultrafine powders of Eudragit^®^ EPO and Opadry^®^ AMB. The results of the morphology study indicated that the coating film was smooth, continuous, and with an excellent uniformity. Dissolution tests and stability studies demonstrated that both Eudragit^®^ EPO- and Opadry^®^ AMB-coated ibuprofen tablets had an instant release profile with an acceptable moisture barrier ability.

Vapor phase deposition method [78,79] is a recently developed coating technology for pharmaceutical solid dosages. It is capable of synthesizing polymer coating films with an engineered surface, functionalities, and topography with satisfactory uniformity. There are three major vapor phase deposition methods in terms of synthesis mechanism and applicability to pharmaceutical environment, namely, initiated chemical vapor deposition (iCVD), plasma enhanced chemical vapor deposition (PE-CVD), and atomic/molecular layer deposition (ALD/MLD). The present dry coating method is a very promising alternative for coating solid dosages, especially for in drug encapsulations.

## 5. Summary and Conclusions

Moisture protection of oral dosages is central to providing stability to the dosages through their shelf life and to ensure the required qualities and desired efficacy especially for those active ingredients with high moisture sensitivity which are easily hydrolyzed. Film coating remains the preferred and most commonly used approach to efficiently achieve moisture barrier function. Both formulation design and process control play a key role in obtaining a qualified coating film.

Normally, a moisture barrier coating formulation is composed of film-forming polymers, plasticizers, and necessary pigments. Film-forming polymers include water-soluble, cationic, anionic, or neutral insoluble polymers from different chemical structures. Several new formulations comprising plant-based materials, such as zein and abietic acid, have been designed and developed. New and effective additives, such as suberin fatty acids (SFAs), have also been developed to improve the commonly used moisture barrier polymers, such as HPMC.

Currently, moisture barrier coating can be performed either by organic solvent coating or by aqueous coating, both of which involve spraying of the coating solution or dispersion onto the surface of the dosages, followed by a drying and curing step to allow film formation. The organic solvent or water in the sprayed coating solution or dispersion needs to be dried immediately after transporting the coating polymers and other excipients onto the surface of the dosages, otherwise surplus organic solvent or water will accumulate and can possibly get into the core, leading to a very soft, sticky, and unstable film. Hence, a careful balance needs to be reached between the spraying speed and the drying rate for a sufficient coat thickness and for a flexible and dry coat. Unlike organic solvent coating, aqueous coating does not have toxicity and environmentally related problems, but it still possesses many limitations such as a longer processing time and much higher energy consumption than solvent coating. Most importantly, aqueous coating is not appropriate for the moisture-sensitive drugs, which is a big challenge for moisture barrier coating. Many efforts have been made to develop dry coating to minimize and eliminate the drawbacks of organic solvent coating, as well as aqueous coating. These dry coating processes include compression coating, hot-melt coating, heat dry coating, electrostatic dry powder coating, and vapor phase deposition method, all of which are capable of forming a qualified moisture barrier film coating. Although each of these dry coatings has its own advantages, they do have many limitations, hindering their applications in pharmaceutical industry. Compared to other dry coating technologies, electrostatic dry powder coating has possibly gained more attention due to its distinguished advantages, such as being highly valued for energy savings and significant reduction of overall operation cost. Most importantly, electrostatic powder coating is capable of being applied for most of the commercialized coating polymers without any selectivity, which make it promising as a green alternative to organic solvent coating or aqueous coating [80].

Despite the success of moisture barrier coating with designed formulations and developed coating processes, further research still needs to be conducted, focusing on some specific drugs such as traditional Chinese medicines (TCMs), due to their extreme sensitivity to moisture.

## Figures and Tables

**Figure 1 pharmaceutics-11-00436-f001:**
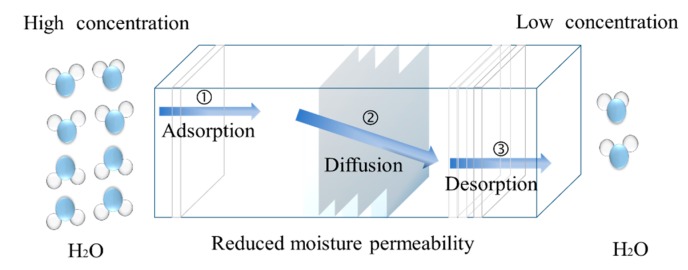
Schematic representation of the moisture uptake.

**Figure 2 pharmaceutics-11-00436-f002:**
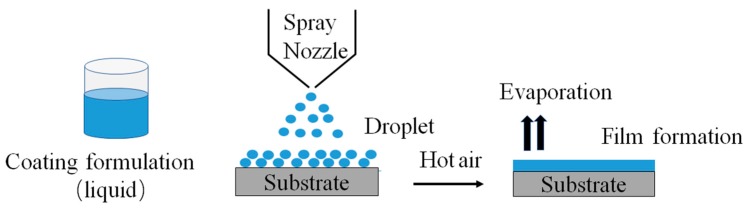
Schematic illustration of the film formation process.

**Figure 3 pharmaceutics-11-00436-f003:**
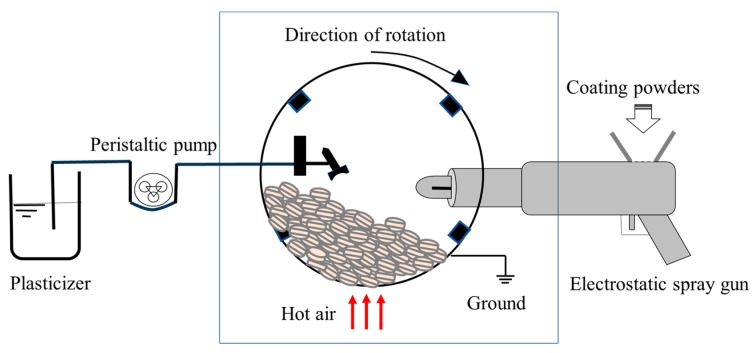
Schematic diagram of the electrostatic powder coating system.

**Table 1 pharmaceutics-11-00436-t001:** Polymers used for moisture barrier film coating.

Type	Trademark	Polymer	Manufacturer
Water-soluble polymers	Opadry^®^ AMB	Polyvinyl alcohol (PVA)	Colorcon (Harleysville, PA, USA)
	Methocel^®^ E3/E5/E6/E15	Hydroxypropyl methyl cellulose (HPMC)	Dow Chemical (Midland, MI, USA)
	Walocel^®^ HM 3 PA/HM 5 PA/HM 6 PA/HM 15 PA		Dow Wolff Cellulosics (Mitterland, MI, USA)
	Pharmacoat^®^ 603/606/615/645		Shin-Etsu (Tokyo, Japan)
	Sepifilm^®^ LP		Seppic (Castres Cedex, France)
	Oxycellulose, Natrosol	Hydroxyethyl cellulose (HEC)	Ashland Aqualon (Covington, Kentucky, USA)
	Kollicoat^®^ IR Protect	Polyvinyl alcohol–polyethylene glycol (PVA–PEG copolymer)	BASF (Ludwigshafen, Germany)
	Kollicoat^®^ IR AquaPolish^®^	Polyvinyl alcohol–polyethylene glycol (PVA–PEG)	BASF (Ludwigshafen, Germany) BioGrund
	Kollicoat^®^ Smartseal 30D	Methyl methacrylate and diethylamino–ethyl ethacrylate copolymer dispersion	BASF (Ludwigshafen, Germany)
	Klucel™	Hydroxypropyl cellulose (HPC)	Ashland (Covington, Kentucky, USA)
Insoluble polymers	Kollicoat^®^ SR 30 D	Polyvinyl acetate	BASF (Ludwigshafen, Germany)
	Auqacoat^®^ ECD	Ethyl cellulose	FMC (Philadelphia, PA, USA)
	Surelease^®^ (Fertigprodukt)		Colorcon (Harleysville, PA, USA)
	Ethocel^TM^		Dow Chemical (Mitterland, MI, USA)
	Eastman CA	Cellulose acetate	Eastman (Rochester, MN, USA)
	Eudragit^®^ RL/ RS 30 D Eudragit^®^ RL/ RS 12.5 Eudragit^®^ RL/ RS 100 Eudragit^®^ RL/ RS PO	Ammonio methacrylate	Evonik (Essen, Germany)
	Aquapolish^®^ R	Ammonio methacrylate copolymer (type A and type B)	Biogrund (Hünstetten, Germany)
	Eudragit^®^ NE 30 D Eudragit^®^ NM 30 D	Poly (ethyl acrylate–co-methyl methacrylate) 2:1	Evonik (Essen, Germany)
Entero-soluble polymers	SSB 55 Pharma	Shellac	Chineway (Shanghai, China)
	Aquacoat^®^ CPD	Cellulose acetate phthalate (CAP)	FMC (Philadelphia, PA, USA)
	Eastman C-A-P NF		Eastman (Rochester, MN, USA)
	CAB Eastman	Cellulose acetate butyrate (CAB)	Eastman (Rochester, MN, USA)
	Eudragit^®^ L30D-55/ L 100-55	Methacrylic acid copolymer, Type A	Evonik (Essen, Germany)
	Eastacryl 30 D NF		Eastman (Rochester, MN, USA)
	Kollicoat^®^ MAE 30 DP/100 P		BASF (Ludwigshafen, Germany)
	Eudragit^®^ L 12.5/ L 100	Methacrylic acid copolymer, Type B	Evonik (Essen, Germany)
	Eudragit^®^ S 12.5/ S 100	Methacrylic acid copolymer, Type C	Evonik (Essen, Germany)
	Eudragit^®^ FS 30 D	Methacrylic acid copolymer	Evonik (Essen, Germany)
	Kollicoat^®^ Smartseal 30 D	Amino diethyl–methacrylate copolymer	BASF (Ludwigshafen, Germany)
	Eudragit^® ®^E/ E 12.5 Eudragit^®^ E PO	Amino dimethyl methacrylate copolymer	Evonik (Essen, Germany)
	Aquapolish^®^ E	Acrylic acid copolymer	Biogrund (Hünstetten, Germany)
	Keltone LV CR	Sodium alginate	FMC (Philadelphia, PA, USA)
	Akucell	Carboxymethyl cellulose CMC	Ashland Aqualon (Covington, Kentucky, USA)

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
