# Peer review of "An Update of Moisture Barrier Coating for Drug Delivery"

_pharmaceutics, 2019, doi:10.3390/pharmaceutics11090436_

Round 1

Reviewer 1 Report

The review paper presents an extensive summary on the moisture protection drug coatings. It is clearly written and concise. I have only the following minor questions, which maybe could add some more useful info in the paper.

1) In chapter 3.1 a discussion about the coating lifetime should be included. Do water-soluble polymers employed as moisture barrier coating have a limited time span for providing good barrier properties in normal ambient conditions? I imagine the normal ambient humidity could degrade the barrier polymer. How does the coating lifetime compare to the water-insoluble coatings?

2) Chapter 4.1: why the organic solvent coating is the preferred coating technology? The advantages of this approach could be more clearly stated.

3) Chapter 4.3: among dry coatings techniques, depositions from the vapor phase should be mentioned. Recently a comprehensive review has been published on the topic: Perrotta et al. Adv. Eng. Mater. 2017, 1700639. In particular, ref 76 focuses on the role of the vapor deposited coating for the API protection from the environment.

Author Response

Reviewer 1:

The review paper presents an extensive summary on the moisture protection drug coatings. It is clearly written and concise. I have only the following minor questions, which maybe could add some more useful info in the paper.

1) In chapter 3.1 a discussion about the coating lifetime should be included. Do water-soluble polymers employed as moisture barrier coating have a limited time span for providing good barrier properties in normal ambient conditions? I imagine the normal ambient humidity could degrade the barrier polymer. How does the coating lifetime compare to the water-insoluble coatings?

Authors’ response: the coating film formed by the water – soluble polymers has a relatively shorter lifetime compared to the film formed by the water-insoluble polymers owing to the degradation of the coatings caused by the ambient humidity during storage. Those has been discussed in the revised manuscript.

2) Chapter 4.1: why the organic solvent coating is the preferred coating technology? The advantages of this approach could be more clearly stated.

Authors’ response: Organic solvent coating still remains the preferred coating method, just for special dosages such as osmotic drug delivery systems and for some specific drugs, especially moisture sensitive drugs. For those special dosages including osmotic drug delivery systems, generally the coating polymers have a very high Tg, which is unacceptable for the aqueous coating to form a uniform and continuous coating film. Additionally for those moisture sensitive drugs, the humidity during the aqueous coating process will cause serious drug degradation. Those information has been enriched in the revised manuscript.

3) Chapter 4.3: among dry coatings techniques, depositions from the vapor phase should be mentioned. Recently a comprehensive review has been published on the topic: Perrotta et al. Adv. Eng. Mater. 2017, 1700639. In particular, ref 76 focuses on the role of the vapor deposited coating for the API protection from the environment.

Authors’ response: dry coated film from vapor phase deposition has been discussed in the Chapter 4.3 accordingly.

Reviewer 2 Report

The paper is quite well written, but it needs further corrections to be published as a review, as follows.

In Introduction: a) Are enumerated the reaction which affect the API stability, but the light is not a chemical reaction. Moreover, the chemical reactions don't "makes the API unstable" but they affect the API therapeutic properties. These should be corrected. b) further was staed that: "Film coating also protects the cores from the external environment preventing oxidation, and blocking light", I think correct is ". Film coating also protects the cores from the external environment blocking light and so preventing oxidation", being known that usually the light accelerate the oxidation.

In "2.1. Mechanism of moisture uptakes", the sentence "As can be seen, the diffusion of water through the film is dependent of the dissolution of the water in the film, and the diffusion of water through the film. " should be rewritten, its meaning is not clear.

In "2.2. Testing of moisture uptake", a) carefully revise the description of the water vapor permeability test, as in this form the procedure is not clear. b) regarding the thermal analysis technique, for DSC, the Tg evolution - water amount relationship should be discussed. c) data about the use of TGA for the estimation of the moisture resistance capacity should be provided.

In "3.3. Pigments " a) references should be provided; b) methods for a better distribution of pigments should be provided; c) examples for use of pigments in coatings should be provided.

In "4. Coating process" a) " The solvent serves as a polymeric system
to dissolve/disperse the polymer " - should be rewritten.

In "4.1. Organic solvent coating" a) " to move with more ease"; " to dissolve most sustained-release" - should be reformulated.

In "4.2. Aqueous polymeric film coatings", a) a reference for " a higher solid content (above 12%, w/w) has also been suggested" should be provided.

In "4.3. Dry coatings" clearly provide the procedure of coating for each method. For electrostatic coating process it is not clear at all.

Generally speaking, as the paper is intended to be a review, more literature data (especially the recent one) should be taken into consideration. Patents should be considered also.

The references should be carefully revised to respect a unique format; reference 1 was repeated as reference 5.

Author Response

The paper is quite well written, but it needs further corrections to be published as a review, as follows.

In Introduction: a) Are enumerated the reaction which affect the API stability, but the light is not a chemical reaction. Moreover, the chemical reactions don't "makes the API unstable" but they affect the API therapeutic properties. These should be corrected. b) further was staed that: "Film coating also protects the cores from the external environment preventing oxidation, and blocking light", I think correct is ". Film coating also protects the cores from the external environment blocking light and so preventing oxidation", being known that usually the light accelerate the oxidation.

Authors’ response: this part has been corrected according to the reviewer’s comment.

In "2.1. Mechanism of moisture uptakes", the sentence "As can be seen, the diffusion of water through the film is dependent of the dissolution of the water in the film, and the diffusion of water through the film." should be rewritten, its meaning is not clear.

Authors’ response: this statement has been revised according to the reviewer’s comment.

In "2.2. Testing of moisture uptake", a) carefully revise the description of the water vapor permeability test, as in this form the procedure is not clear. b) regarding the thermal analysis technique, for DSC, the Tg evolution - water amount relationship should be discussed. c) data about the use of TGA for the estimation of the moisture resistance capacity should be provided.

Authors’ response: those parts have been carefully revised and enriched according to the reviewer’s comments.

In "3.3. Pigments" a) references should be provided; b) methods for a better distribution of pigments should be provided; c) examples for use of pigments in coatings should be provided.

Authors’ response: we have re-written this part according to the reviewer’s comments.

In "4. Coating process" a) " The solvent serves as a polymeric system to dissolve/disperse the polymer " - should be rewritten.

Authors’ response: here the “solvent” refers to either organic solvent or water, which are used to dissolve or disperse the coating polymer and other additives. This part has been revised according to the reviewer’s comments.

In "4.1. Organic solvent coating" a) " to move with more ease"; " to dissolve most sustained-release" - should be reformulated.

Authors’ response: this statement has been revised.

In "4.2. Aqueous polymeric film coatings", a) a reference for " a higher solid content (above 12%, w/w) has also been suggested" should be provided.

Authors’ response: the related reference has been provided in the revised manuscript.

In "4.3. Dry coatings" clearly provide the procedure of coating for each method. For electrostatic coating process it is not clear at all.

Authors’ response: the electrostatic coating process has been described with details and figure of the coating systems has been provided in the revised manuscript.

Generally speaking, as the paper is intended to be a review, more literature data (especially the recent one) should be taken into consideration. Patents should be considered also.

Authors’ response: we believe that the original manuscript had already included lots of recent and related literatures. Anyway, in order to make it better, we have enriched the manuscript with more recent literature data (such as a new dry coating technology: the vapor-phase deposition method), and also several related patents have been referred.

The references should be carefully revised to respect a unique format; reference 1 was repeated as reference 5.

Authors’ response: references have been carefully modified.

Reviewer 3 Report

In this review Yang and coauthors reports on application of moisture-barrier film coating for drug delivery.

In my opinion, since the declared purpose of this work is to provide the reader with updates on the design and realization of the moisture barrier film, the authors should integrate and update the bibliographic references in the light of the most recent publications. Even the division of the paragraphs could be improved to the advantage of greater clarity. For this reason I recommend minor revision.

Comments:

page 1, Introduction section: this paragraph should be improved and made more schematic, as well as the rest of the work. An interesting reading for the authors could be Excipient Selection in Oral Solid Dosage Formulations Containing Moisture Sensitive Drugs (Ali R. Rajabi-Siahboomi et al.) in Excipient Applications in Formulation Design and Drug Delivery edited by Ajit S. Narang e Sai H. S. Boddu, a book from 2016 that is incredibly not mentioned in this work;

page 1, line 36: replace “light, thermal degradation” with “photo and thermal degradation”;

page 2, line 61: add an appropriate reference;

page 3, lines 106-120: this paragraph is confused and hasty, it should be made clearer and enriched with bibliographical references to which the reader can refer for further information;

page 4, lines 139-152: the authors argue about the thickness of the coating; this introductory part should be integrated in the following paragraphs or accompanied by appropriate bibliographical references;

page 5: Table 2 should be table 1 since it is the only table;

page 5: the table relating to water-soluble polymers could be completed by also adding data relating to water-insoluble polymers and entero-soluble polymers. The authors keep in mind that there are more products on the market than those mentioned in the work, and therefore they should enrich the discussion. Alternatively they could add two more tables rather than modify this. In any case, for a quick and effective consultation it would be useful to add a column with the appropriate bibliographic references;

page 5, line 177: recently with this purpose the Eudragit S100 polymer has been used to cover liposomes containing curcumin in polymeric cluster using a pH-driven and organic solvent-free process (De Leo et al., Encapsulation of curcumin-loaded liposomes for colonic drug delivery in a pH-responsive polymer cluster using a pH-driven and organic solvent-free process, Molecules, 2018).

page 5, lines 200-215: again, a whole paragraph dedicated to pigments that does not refer to any bibliographic reference

pages 6-7 lines 230-253: a paragraph so long is devoid of any bibliographic reference, please check.

Author Response

In this review Yang and coauthors reports on application of moisture-barrier film coating for drug delivery.

In my opinion, since the declared purpose of this work is to provide the reader with updates on the design and realization of the moisture barrier film, the authors should integrate and update the bibliographic references in the light of the most recent publications. Even the division of the paragraphs could be improved to the advantage of greater clarity. For this reason I recommend minor revision.

Comments:

page 1, Introduction section: this paragraph should be improved and made more schematic, as well as the rest of the work. An interesting reading for the authors could be Excipient Selection in Oral Solid Dosage Formulations Containing Moisture Sensitive Drugs (Ali R. Rajabi-Siahboomi et al.) in Excipient Applications in Formulation Design and Drug Delivery edited by Ajit S. Narang e Sai H. S. Boddu, a book from 2016 that is incredibly not mentioned in this work;

Authors’ response: the current review focused on the recent updates of coating formulation development and process design for the function of moisture barrier, the influence of the core formulation including APIs and excipients are not the main purpose. But anyway, a small discussion on the influence of excipient has been added in the revised manuscript in the “Introduction” part.

page 1, line 36: replace “light, thermal degradation” with “photo and thermal degradation”;

Authors’ response: this has been corrected accordingly.  

page 2, line 61: add an appropriate reference;

Authors’ response: an appropriate reference has been added accordingly.

page 3, lines 106-120: this paragraph is confused and hasty, it should be made clearer and enriched with bibliographical references to which the reader can refer for further information;

Authors’ response: this part has been revised accordingly, we believe this revision make it more clearer.

page 4, lines 139-152: the authors argue about the thickness of the coating; this introductory part should be integrated in the following paragraphs or accompanied by appropriate bibliographical references;

Authors’ response: we believe keeping this part in the current position is better, but we have enriched it by adding many related bibliographical references. 

page 5: Table 2 should be table 1 since it is the only table;

Authors’ response: it has been corrected accordingly.

page 5: the table relating to water-soluble polymers could be completed by also adding data relating to water-insoluble polymers and entero-soluble polymers. The authors keep in mind that there are more products on the market than those mentioned in the work, and therefore they should enrich the discussion. Alternatively they could add two more tables rather than modify this. In any case, for a quick and effective consultation it would be useful to add a column with the appropriate bibliographic references;

Authors’ response: considering the reviewer’s suggestion, which is very reasonable, we have enriched the Table by including all the three major coating polymers: water-soluble polymers, water-insoluble polymers and entero-soluble polymers.

page 5, line 177: recently with this purpose the Eudragit S100 polymer has been used to cover liposomes containing curcumin in polymeric cluster using a pH-driven and organic solvent-free process (De Leo et al., Encapsulation of curcumin-loaded liposomes for colonic drug delivery in a pH-responsive polymer cluster using a pH-driven and organic solvent-free process, Molecules, 2018).

Authors’ response: this specific reference has been included in the revised manuscript accordingly.

page 5, lines 200-215: again, a whole paragraph dedicated to pigments that does not refer to any bibliographic reference

Authors’ response: several related references have been included in the revised manuscript.

pages 6-7 lines 230-253: a paragraph so long is devoid of any bibliographic reference, please check.

Authors’ response: according to this comment, appropriate references have been included in the revised manuscript.

Round 2

Reviewer 2 Report

The authors nicely reviewed their paper, which is more appropriate for publication now.